# Identification and Expression Profiling of *TGA* Transcription Factor Genes in Sugarcane Reveals the Roles in Response to *Sporisorium scitamineum* Infection

**Zhengying Luo [1,2], Xin Hu [2], Zhuandi Wu [2], Xinlong Liu [2], Caiwen Wu [2,*] and Qianchun Zeng [1,*]**

[1] College of Agronomy and Biotechnology, Yunnan Agricultural University, Kunming 650201, China
[2] Yunnan Key Laboratory of Sugarcane Genetic Improvement, Sugarcane Research Institute, Yunnan Academy of Agricultural Sciences, Kaiyuan 661699, China
* Correspondence: gksky_wcw@163.com (C.W.); zqch@yahoo.com.cn (Q.Z.)

**Abstract:** TGA transcription factor (TF) family genes play a major role in the regulation of plant growth and development as well as in the defense against pathogen attack. Little is known about the *TGA* family genes and their functions in sugarcane. Here, a total of 16 *TGA* members were identified in the sugarcane genome by bioinformatic approaches. All members exhibited similar conserved motifs and contained a bZIP domain and a DOG1 domain, except for *ShTGA15/16*. Phylogenetic analysis demonstrated that 16 *ShTGA* family genes could be divided into eight clades, and evolved differently from *Arabidopsis TGAs*. All *ShTGA* family genes suffered a purifying selection during evolution. A wide range of cis-regulatory elements were found in the promoter of *ShTGA* genes including hormone regulatory elements, adversity response elements, light responsive elements, and growth and development regulatory elements. Most *ShTGA* expressions were increased in bud growth and developmental processes except for *ShTGA10/11*. It is worth noting that the expression of *ShTGA13* was decreased after sugarcane was infected with *Sporisorium scitamineum*, and it was highly expressed in the resistant variety compared to the susceptible variety. Adding IAA, GA$_3$ and SA restored the expression of *ShTGA13*, suggesting an association with plant hormone regulatory pathways. Our study provides a framework for further functional studies of important *ShTGA* genes in development and stress response, and uncovered a previously unrecognized role of *ShTGA13* in regulating resistance against *S. scitamineum*.

**Keywords:** sugarcane; TGA transcription factor; gene expression; evolutionary analysis; *Sporisorium scitamineum*

## 1. Introduction

TGACG-Binding (TGA) transcription factors (TFs) are the subfamily of basic region/leucine zipper (bZIP) TFs which are extensively present in all eukaryotes. These TGAs act at the interface between the DNA and the regulatory proteins by binding to cis-regulatory elements with TGACG (also called activation sequence-1, as-1) [1,2]. The first plant TGA transcription factor was tobacco *TGA1a*, characterized in 1989 [3]. In 1992, 10 TGA members were discovered and were divided into five clades in *Arabidopsis* [4]. Identification of the TGA transcription factor family in other plant species has been reported recently [5–9]. The function and regulatory mechanism of TGAs have been well studied in *Arabidopsis TGAs* mutants, revealing their importance in a wide range of biological processes. ATTGA1/2/3/4/5/6/7 were found to constitutively interact with the *non-repressor of pathogenesis-related gene 1* (*NPR1*), which is a key positive regulator of the salicylic acid (SA)-dependent signaling pathway [10–12]. Thus, these ATTGAs played the essential role of inducing *pathogenesis-related* (*PR*) *1* expression and subsequently SAR activation in response to pathogen attack [13,14]. ATTGA2/5/6 were believed to be essential activators of jasmonic acid/ethylene-induced defense responses [15,16]. ATTGAs were

also found to be involved in regulating the detoxification pathway [17], UV-B stress [18], $Cr^{6+}$ tolerance [19], drought resistance [20] and SA-induce redox state [21]. Several studies showed that the expression of *TGAs* was differentially regulated subsequent to pathogen infection and abiotic stress in plants [22–24]. Their regulatory roles were usually connected to plant hormonal pathways and they could affect plant immunity by modulating the basal promoter activity of the *PR-1* gene [25,26]. Beyond their importance for biotic and abiotic stress responses, the *TGA* gene family is associated with plant growth and development including shoot apical meristem (SAM) maintenance, flowering, inflorescence architecture development, root growth and circadian rhythm [27–30].

Sugarcane (*Saccharum* spp. hybrid) is the world's largest tropical and subtropical crop constituting the chief source of sugar. It has a long growth season, and is attacked by many fungal pathogens. Sugarcane smut, caused by *Sporisorium scitamineum*, is the most challenging fungous disease. It was first reported in Natal province, South Africa, in 1877, and is now widely prevalent in major sugarcane planting areas around the world. To cope with the smut, sugarcane developed various mechanisms to impair pathogen colonization, proliferation and spread [31]. It was documented that the progression of *S. scitamineum* infection was accompanied by distinct gene transcriptional changes in plant hormone biosynthesis and signal transduction [32–34]. Auxin, gibberellin (GA), abscisic acid (ABA), ethylene (ETH), jasmonic acid (JA) and SA were more apparent in response to smut fungus invasion and played complex roles in regulating plant defense responses in cooperative or antagonistic mode [33–38]. The regulatory mechanisms of these hormones are still unspecified, some of which achieved varied roles in different circumstances. Studies on the roles of plant hormones in the interaction between sugarcane and smut fungus are still lacking.

The activity of TGA was connected to different plant hormone signaling pathways including auxin, GA, ABA, JA and SA [38–43]. The research suggests a role of the plant hormone signaling pathway in the defense of sugarcane smut, revealing a potential mechanism by which sugarcane TGA transcription factors mediated appropriate adjustment of gene expression in hormone signal transduction cascades. However, little is known about sugarcane TGA family members and their roles in plant developmental processes and pathogens defense. In this study, we identified the sugarcane TGA family members at the genome level and investigated the physico-chemical properties, gene structure, molecular evolution, and promoter elements. Furthermore, we analyzed the expression of sugarcane *TGA* family genes in response to *S. scitamineum* infection combined with different plant hormone treatments. Our finding suggests a unique role of *ShTGA13* in sugarcane–smut interaction, which can be further explored in explaining the smut-resistant mechanism.

## 2. Materials and Methods

### 2.1. Materials

The study was conducted on a smut-susceptible sugarcane genotype, ROC22 and a smut-resistant sugarcane genotype, YZ05-51 [44], which were produced by the China National Germplasm Repository of Sugarcane, Kaiyuan, China. Robust stems were selected and cut into single bud setts, which were then immersed under flowing water for 24 h at room temperature so as to remove dirt and microorganisms. The setts were dried at room temperature and were transferred to the light incubator (12 h light-dark cycle, 32°C, 80% humidity) for germination. The source of *S. scitamineum* inoculum was collected from the main cultivar ROC22 at Kaiyuan City, Yunnan Province, China in 2021, and was stored at 4 °C. The germination capacity of the teliospores of *S. scitamineum* was checked in 1% water agar; teliospores that showed over 90% germination were used for inoculation.

### 2.2. S. scitamineum Inoculation and Applied Treatments

After three days of germination, sugarcane buds were injected with *S. scitamineum* inoculum via puncture as described previously [33]. Five different treatments and at least forty setts per treatment were used including control (inoculated with sterile water), sugarcane smut infected plants, and infected plants supplemented with plant hormones (IAA, GA$_3$, and SA were used separately). The specific phytohormones and their concentrations were $2 \times 10^{-3}$ M IAA, $1 \times 10^{-3}$ M gibberellin (GA$_3$), $5 \times 10^{-3}$ M SA referring to previous studies [45,46]. The treated bud setts continued to culture in the light incubator, and the shoot apical meristem samples were collected at 0, 1, 3 and 7 d after inoculation. At least two bud setts were mixed into one sample, and three biological replicates were adopted at each condition. Samples were frozen immediately in liquid N$^2$ for further analysis.

### 2.3. Sequence Retrieval

The whole-genome shotgun contigs database of smut-resistant sugarcane cultivar SP80-3280 (txid: 193079, accession number in PRJNA431722) was obtained from the National Center for Biotechnology Information (NCBI), and the genomic data of *S. spontaneum* AP85-441, a wild *Saccharum* species, were also retrieved (accession number in QVOL00000000). The TGA nucleotide sequences and amino acid sequences of *Oryza sativa* Indica Group (txid: 39946) and *Arabidopsis thaliana* (L.) Heynh (txid: 3702) were obtained by querying the reported *TGA* gene name [42,47] from the *Arabidopsis* Information Resource and the Rice Genome Annotation Project (http://rice.plantbiology.msu.edu/, accessed on 3 July 2022), respectively. The gene IDs of *Arabidopsis TGA* were *ATTGA1* (At5g65210), *ATTGA2* (At5g06950), *ATTGA3* (At1g22070), *ATTGA4* (At5g10030), *ATTGA5* (At5g06960), *ATTGA6* (At3g12250), *ATTGA7* (At1g77920), *ATTGA8* (At1g68640), *ATTGA9* (At1g08320), and *ATTGA10* (At5g06839). The gene IDs of rice *TGA* were *OSTGA1* (Os05t0443900), *OSTGAP1* (Os04t0637000), *OSTGA2* (Os01t0808100), *OSTGA3* (Os03t0318600), *OSTGA5* (Os01t0279900), *OSTGA10* (Os09t0489500), *OSTGAL11* (Os12t0152900), *OSbZIP47* (Os06t0265400) and *OSbZIP49* (Os06t0614100).

### 2.4. Screening and Identification of TGA Family Genes in Sugarcane

*ATTGA* family genes and *OSTGA* family genes were used as query sequences to search for sugarcane *TGA* family genes in the genome database of *Saccharum* spp. hybrid SP80-3280 and *S. spontaneum* AP85-441 (downloaded from NCBI) through the Basic Local Alignment Search Tool and default parameters [48]. The resulting hits were filtered by E-value ($1 \times 10^{-8}$), and only the longest sequence was retained if several results were found for the same gene. The open reading frames were sought by the Open Reading Frame Finder program (https://www.ncbi.nlm.nih.gov/orffinder/, accessed on 3 July 2022). The conserved domain analysis was performed and checked by the NCBI-Conserved Domain Database [49] and PFAM protein family database (http://pfam.xfam.org/, accessed on 3 July 2022) [50]. All sugarcane TGA protein sequences were analyzed by ExPASy to obtain their basic physical and chemical properties, such as molecular weight (MW), isoelectric point (pI), amino acid composition, instability coefficient and total average hydrophilicity (https://www.expasy.org/protparam/, accessed on 3 July 2022) [51]. The protein subcellular localization was predicted by CELLO v.2.5 [52].

### 2.5. Analysis of Gene Structure and Conserved Motifs

The homology and structural characteristics of all ShTGA family members were conducted by BioEdit v7.0 with the default parameters [53]. Gene structures of *ShTGA* genes were analyzed and visualized by TBtools [54], and the motif structures in the ShTGA protein sequences were analyzed by the MEME program (Version 5.1.1, University of Nevada, Reno) with a maximum number of motifs set to 10 [55]. Letters that appeared in each position constituted a position-specific probability matrix, which could be used to judge the possible motifs in the sequence group.

### 2.6. Phylogenetic Analysis of ShTGA Proteins and Calculation of Ka/Ks

The alignment of multiple amino acid sequences of the selected 35 TGA family members, composed of 10 *Arabidopsis* TGAs, 9 rice TGAs and 16 sugarcane TGAs, was performed using MUSCLE in MEGA 11 software with default parameters [56]. The phylogenetic tree based on the alignments was constructed using the maximum likelihood method with 1000 bootstrap replicates, the Jones–Taylor–Thornton (JTT) model, gamma distribution and partial deletion [56]. The non-synonymous (Ka) and synonymous (Ks) substitution ratios were also calculated using Compute Pairwise Distances in MEGA 11, and the divergence time (T) was calculated according to the Ks value by $T = Ks/2\lambda \times 10^{-6}$ Mya ($\lambda = 1.5 \times 10^{-8}$).

### 2.7. Cis-Regulatory Elements Prediction of ShTGA Gene Family

The 2000-bp upstream sequence of the coding region of each *ShTGA* family gene was obtained to investigate cis-regulatory elements. The putative cis-regulatory elements in the promoter sequences were analyzed via PlantCARE (http://bioinformatics.psb.ugent.be/webtools/plantcare/html/, accessed on 3 July 2022) and visualized by Microsoft Excel version 2010.

### 2.8. Experimental Validation of ShTGA Gene Expression Levels by qRT-PCR

The shoot apical meristem samples were used to extract RNA using the FastPure Plant Total RNA Isolation Kit (Vazyme Biotech Co., Ltd., Nanjing, China), and the RNA was reverse transcribed into cDNA using HiScript III RT SuperMix for qPCR (Vazyme Biotech Co., Ltd., Nanjing, China). Gene-specific primer synthesis was conducted by the Beijing Genome Institute (Shenzhen, China). The specific nucleotide sequences of primers for quantifying every *ShTGA* expression were listed in Supplementary Table S1. Gene expressions were normalized against an internal reference *GAPDH* gene. The volume of the qRT-PCR reaction was 20 μL, including 10 μL FastStart Universal SYBR Green PCR Master, 0.4 μL primer and 2 μL template cDNA. The qRT-PCR program was set at 95 °C for 2 min, with 40 cycles of 95 °C for 10 s and 60 °C for 30 s. Three independent biological samples were evaluated for every condition and all reactions of each sample were performed in triplicate for the analysis of *ShTGA* gene expression. The relative expression of *ShTGA* genes was calculated using the $2^{-\Delta\Delta c(t)}$ method.

### 2.9. Statistical Analysis

In the qRT-PCR analysis, the differences between relative gene expressions were analyzed using the one-way ANOVA test, and the least significant differences (LSD) method was used for further comparison between two groups at $p < 0.05$ (marked with *) or $p < 0.01$ (marked with **) (SPSS 20.0, Inc., Chicago, IL, USA).

## 3. Results

*3.1. Screening TGA Transcription Factors Family Genes for Basic Physic-Chemical Properties of ShTGA Coding Proteins*

Sugarcane *TGA* family genes, named *ShTGA1* to *ShTGA16*, were identified from the sugarcane genome database by sequence homology blast and domain confirmation. The number of nucleotides of sixteen *TGA* genes ranged from 2172 to 11238, and the sequence identity of their coding DNA sequences varied from 2.6% to 23.7%. The sixteen TGA proteins had from as few as 330 (ShTGA5) to as many as 652 (ShTGA16) amino acids (Table 1). The molecular weights varied from 36.7 to 68.7 kD. The isoelectric points ranged from 5.86 (ShTGA15) to 9.67 (ShTGA13). The bZIP domain was contained in all ShTGA proteins, while the DELAY OF GERMINATION1 (DOG1) domain was presented in fourteen ShTGA proteins except for in ShTGA15 and ShTGA16. The grand average of the hydropathicity of all ShTGA proteins was <0, suggesting that those proteins were hydrophilic. All sixteen ShTGA proteins were classified unstable as they had a high instability coefficient (instability coefficient >40) and a low aliphatic index (<90). Protein subcellular prediction showed that all ShTGA proteins were located in the nucleus.

**Table 1.** Physico-chemical parameters and subcellular predictions of TGA family members in *Saccharum* spp. hybrid.

| Gene ID | Accession Number | Number of Amino Acid | MW | pI | BZIP Domain Location | DOG1 Domain Location | GRAVY | Instability Index | Aliphatic Index | Subcellular Location |
|---|---|---|---|---|---|---|---|---|---|---|
| ShTGA1 | OP205433 | 407 | 45,278.52 | 6.35 | 115–159 | 186–400 | −0.399 | 48.79 | 77.79 | Nuclear |
| ShTGA2 | OP205434 | 375 | 41,336.72 | 6.06 | 84–128 | 151–372 | −0.319 | 51.74 | 80.21 | Nuclear |
| ShTGA3 | OP205435 | 335 | 37,385.27 | 8.92 | 49–93 | 116–332 | −0.533 | 55.77 | 82.54 | Nuclear |
| ShTGA4 | OP205436 | 333 | 37,116.98 | 8.9 | 47–91 | 114–330 | −0.522 | 55.89 | 83.63 | Nuclear |
| ShTGA5 | OP205437 | 330 | 36,652.12 | 7.08 | 44–88 | 111–327 | −0.593 | 58.34 | 79.64 | Nuclear |
| ShTGA6 | OP205438 | 332 | 36,931.58 | 6.68 | 45–89 | 112–329 | −0.560 | 57.99 | 84.52 | Nuclear |
| ShTGA7 | OP205439 | 463 | 50,666.80 | 6.23 | 176–220 | 243–460 | −0.498 | 55.12 | 78.55 | Nuclear |
| ShTGA8 | OP205440 | 485 | 53,170.47 | 6.1 | 179–223 | 243–456 | −0.460 | 56.59 | 72 | Nuclear |
| ShTGA9 | OP205441 | 431 | 47,270.06 | 6.81 | 125–169 | 189–402 | −0.472 | 48.01 | 76.91 | Nuclear |
| ShTGA10 | OP205442 | 527 | 57,470.71 | 6.29 | 216–260 | 281–495 | −0.374 | 58.98 | 77.61 | Nuclear |
| ShTGA11 | OP205443 | 514 | 56,653.49 | 5.91 | 198–240 | 263–477 | −0.460 | 65.57 | 73.04 | Nuclear |
| ShTGA12 | OP205444 | 496 | 55,227.21 | 6.88 | 190–232 | 270–484 | −0.565 | 68.92 | 75.02 | Nuclear |
| ShTGA13 | OP205445 | 306 | 33,014.46 | 9.67 | 145–188 | 207–306 | −0.503 | 50.70 | 72.91 | Nuclear |
| ShTGA14 | OP205446 | 425 | 46,972.51 | 7.87 | 138–181 | 205–415 | −0.393 | 61.96 | 80.64 | Nuclear |
| ShTGA15 | OP205447 | 567 | 60,463.29 | 5.86 | 113–173 | / | −0.382 | 51.12 | 73.69 | Nuclear |
| ShTGA16 | OP205448 | 652 | 68,705.2 | 8.83 | 177–237 | / | −0.449 | 45.38 | 71.27 | Nuclear |

MW, molecular weight; pI, isoelectric point; GRAVY, the grand average of hydropathy.

*3.2. Gene Structures and Protein Domains Analysis of ShTGAs*

Converse motif analysis of the sixteen ShTGA proteins showed that motif numbers ranged from 2 (ShTGA16) to 8 (ShTGA8/9/14) (Figure 1). Motif 2 was present in all ShTGAs and more than 81% of the ShTGA proteins had motif 5, 1, 4, 6 and 3, which were partially absent in ShTGA13 and entirely absent in ShTGA15/16. Only ShTGA8/9 contained motif 10, and only ShTGA15/16 had motif 9. Gene structure analysis suggested that the number of exons of the sixteen *ShTGA* genes ranged from 2 (*ShTGA16*) to 12 (*ShTGA8/10/11/12*). Correspondingly, the number of introns ranged from 1 (*ShTGA16*) to 11 (*ShTGA8/10/11//12*). The maximum number of exons was eight, which was observed in *ShTGA3/4/5/6/13* genes. The exon and intron distribution patterns varied considerably, suggesting that there are function variations among different *ShTGA* genes.

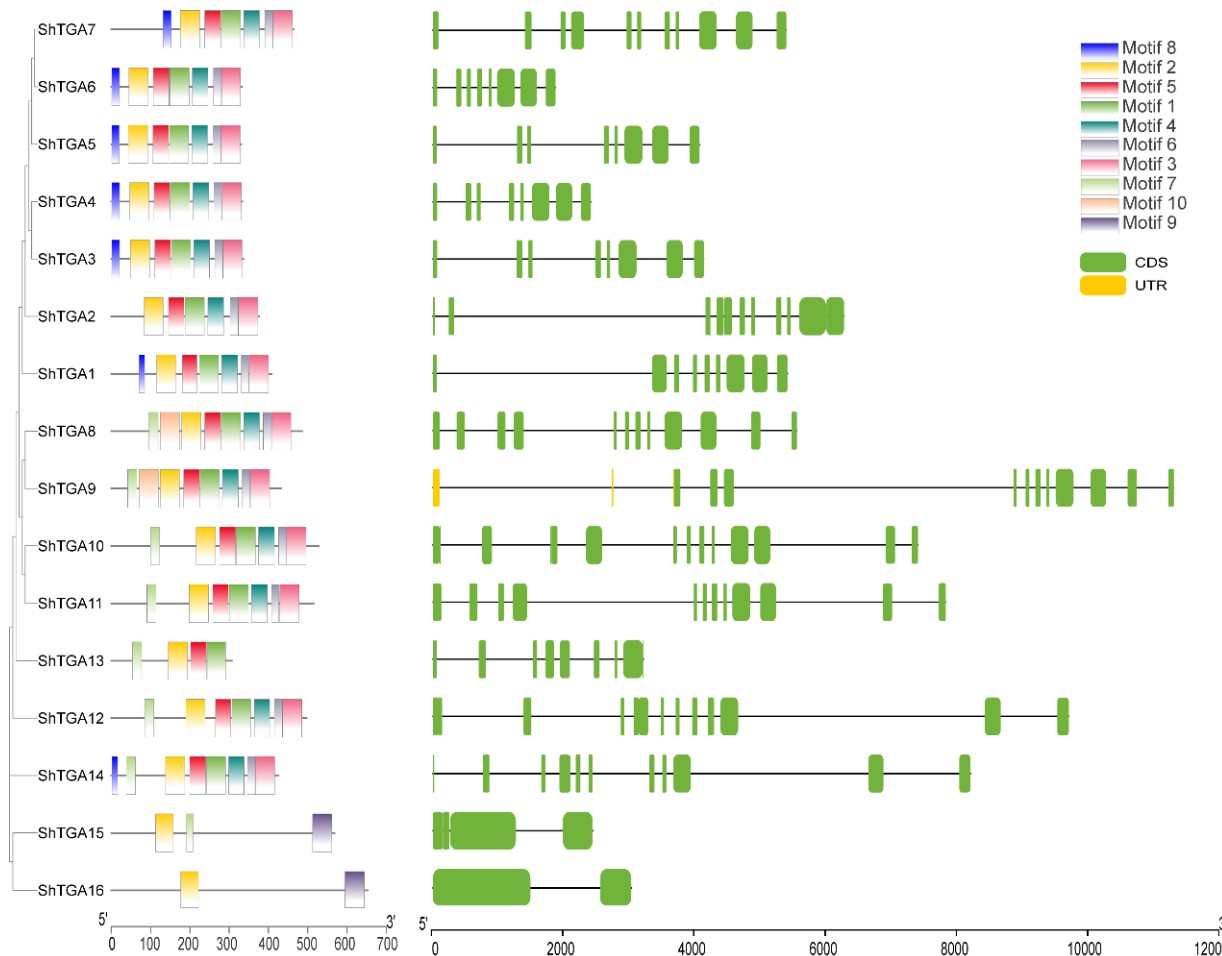

**Figure 1.** Analysis of conserved motifs and gene structure domains in all ShTGA family members. Phylogenetic tree was constructed using the ShTGA protein sequences. Ten types of conserved motifs were predicted, and different motifs are shown in different color boxes. The sequence information for each motif is provided in Supplementary Table S2. The gene structures of *ShTGA* members were visualized; coding sequence (CDS) and untranslated regions (UTR) are shown as light green boxes and yellow boxes, respectively.

### 3.3. Evolution Analysis of ShTGA Family Members

Phylogenetic analysis demonstrated that 35 TGA proteins were separated into eight groups with high bootstrap support (Figure 2). The numbers of sugarcane, *Arabidopsis*, and rice *TGA* genes in each of groups were I (6, 4, 4), II (1, 4, 1), III (2, 0, 1), IV (2, 1, 1), V (1, 1, 1), VI (1, 0, 1), VII (1, 0, 0) and VIII (2, 0, 0), respectively. Group I had the largest number of TGA family members, and TGAs in the same group may have similar functions. The ShTGA proteins were distributed in each group, while *Arabidopsis* and rice TGAs were present in half and three-quarters of the groups. This indicated that species differences might cause the TGA family members to cluster separately.

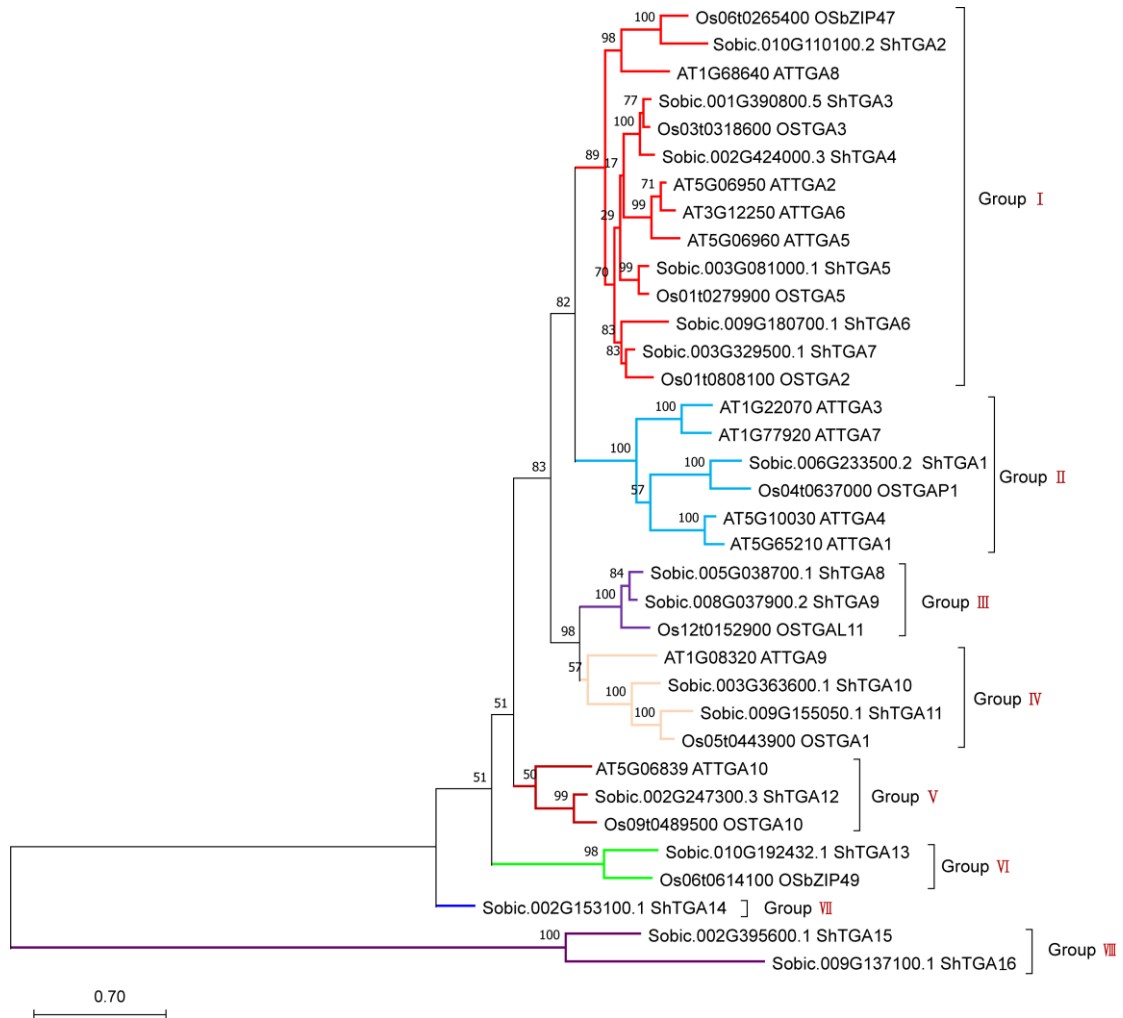

**Figure 2.** Maximum likelihood phylogenetic analysis of TGA families from *Arabidopsis*, rice and sugarcane. Protein sequences were aligned using MUSCLE, and phylogenetic tree reconstruction was made using maximum likelihood method under the best model selection in MEGA11 software with 1000 replicates of rapid bootstrap and LRT statistics. The bootstrap value in which the associated taxa clustered together are shown next to the branches. The scale bar represents 0.07 units of amino acid substitutions per site.

In addition, we analyzed the substitution rate ratio Ka/Ks and divergence time between the *TGA* gene family members (Table S3). The results showed that all Ka/Ks ratios were less than 1, revealing that the evolution of *TGA* family genes was under a purifying selection. The differentiation times of these gene pairs occurred from 25.2 to 105.3 Mya.

### 3.4. Cis-Elements Analysis in ShTGAs Promoter Regions

To better understand the regulatory functions of the *ShTGAs*, the cis-regulatory elements were identified in the 2000 bp upstream promoter sequences of sixteen *ShTGA* genes. The gridding diagram showed that the cis-regulatory elements of the *ShTGA* family genes were divided into four categories: hormone regulatory element, adversity response element, light responsive element, and growth and development regulatory elements (Figure 3). There was the largest number of cis-regulatory elements in the category of hormone regulatory elements including abscisic acid, auxin, ethylene, GA, MeJA and SA responsiveness. The second largest category was adversity response elements, which contained anaerobic induction, stress responsiveness, drought responsiveness, wound-responsive element, etc. The major types of the light responsive element were G-box,

GT1-motif, Box4, Sp1 and GATA-motif. The growth and development regulatory elements mainly had meristem specific activation, seed-specific regulation and zein metabolism regulation. The hormone regulatory elements frequently presented in the promoter regions of all *ShTGA* genes, indicating that these *ShTGAs* might be involved in various phyto-hormone signaling pathways. For individual genes, *ShTGA6* had a considerable number of abscisic acid responsive elements including six ABRE elements, two AAGAA-motif and one ABRE3a. *ShTGA9* contained five TGA-elements which were involved in auxin responsiveness. The MeJA-responsiveness elements were found in the promoter regions of all *ShTGA* genes. Except for *ShTGA4* and *ShTGA13*, all *ShTGA* genes had the TGACG motif, also called the as-1 (activation sequence-1) element, which was usually bound to the SA-induced *PR-1* gene. In addition, *ShTGA* genes were found to contain various adversity response elements, such as anaerobic environment, drought, low-temperature and wound. This suggested that *ShTGA* genes were predicted to be involved in the response to various environmental stimuli.

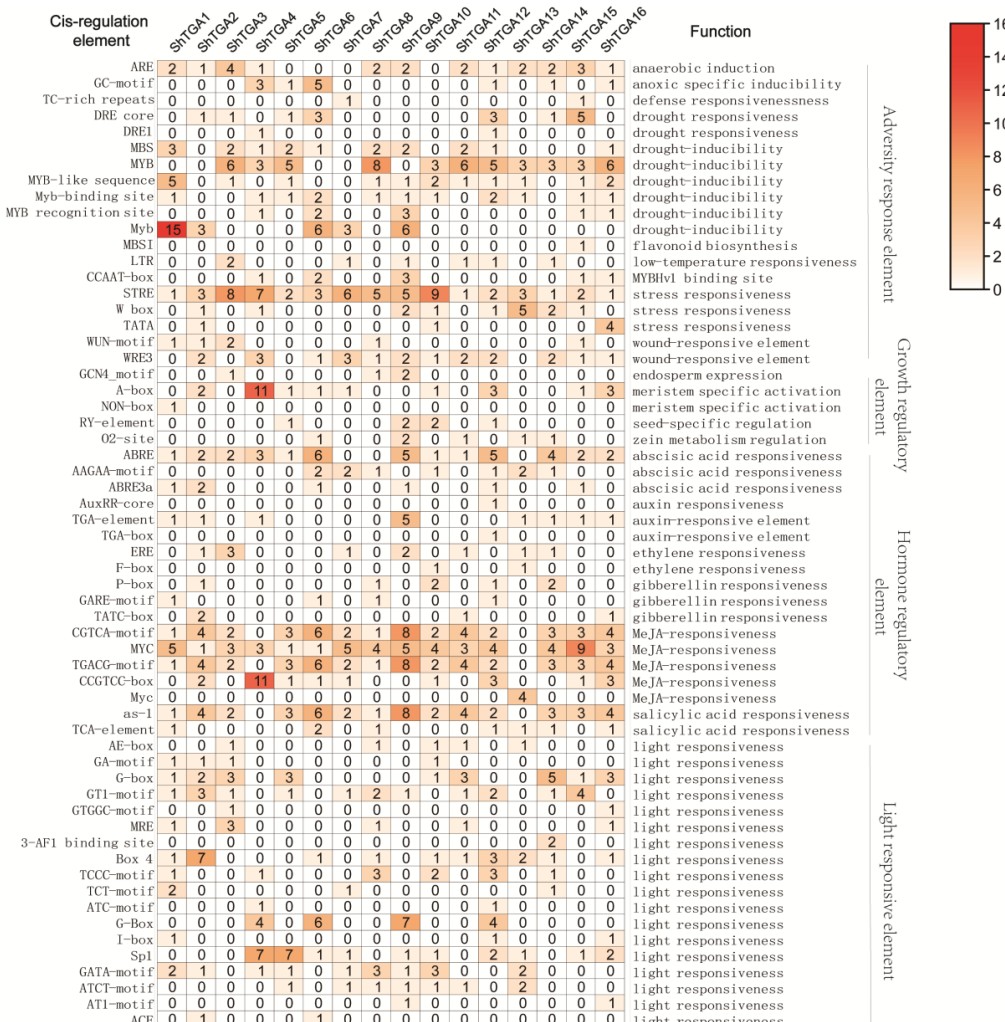

**Figure 3.** Number of varied cis-regulatory elements identified in the promoter regions from *TGA* family genes in sugarcane. Cis-regulatory elements related to hormone response, environmental stress and development were identified by Plant Promoter Analysis Navigator from PlantPAN 3.0 database, using 2000 bp upstream from the translation start site from each gene.

### 3.5. Expression Profile Analysis of ShTGA Genes in the Response to S. scitamineum Infection

Since *TGAs* were originally found for their crucial roles in plant immunity response by coordinated plant hormone signaling pathways, we want to examine whether *ShTGAs* play a role in regulating the smut-resistance of sugarcane accompanied with IAA, GA$_3$ and SA treatment. The expression levels of fifteen *ShTGA* family genes were observed under the infection of *S. scitamineum* as well as when dding hormone, while *ShTGA6* was not detected (Figure 4). Under the condition of inoculation with sterile water (control group), thirteen *ShTGA* family genes, except for *ShTGA10/11*, were shown to be up-regulated during bud growth, indicating their roles in regulating these developmental processes. When sugarcane was inoculated with *S. scitamineum*, there were eleven *ShTGA* family gene expression increases except for *ShTGA10/11/13/14*. Compared with the control treatment, the expressions of *ShTGA13* were inhibited at day3 and day7 after inoculation; the expressions of *ShTGA14* were also decreased at day7 after inoculation. In the circumstance of *S. scitamineum* inoculum supplemented with IAA, it was found that the expressions of twelve *ShTGA* family genes except for *ShTGA2/10/11* were obviously elevated. Apart from *ShTGA2/11*, the other thirteen *ShTGA* family genes exhibited up-regulation in the treatment of *S. scitamineum* inoculum incorporated GA, as well as SA addition. After pathogens inoculation, adding hormone down-regulated the expression of *ShTGA2* but restored the expression of ShTGA 13/14 to control levels. To further investigate the possible role of *ShTGA13* in sugarcane-*S. scitamineum* interaction, the expression of *ShTGA13* was compared between the susceptible (ROC22) and resistant (YZ05-51) varieties (Figure 5). The results showed that the expression of *ShTGA13* was also decreased after ROC22 infected with *S. scitamineum*. It is noteworthy that *ShTGA13* was significantly highly expressed in YZ05-51 compared to ROC22 under all experimental conditions. In general, the expression of *ShTGA13* was regulated subsequent to pathogen infection, and higher expression level was maintained in resistant variety indicating that *ShTGA13* had relevance for the resistance against *S. scitamineum*.

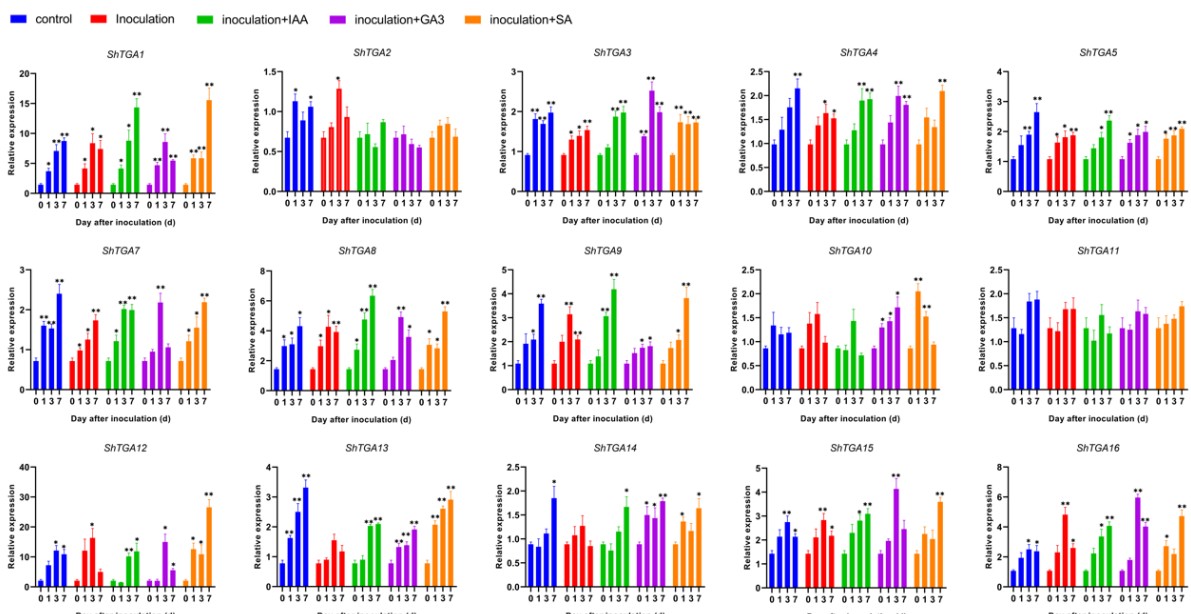

**Figure 4.** Gene expression of ShTGA genes in response to *S. scitamineum* infection supplemented with different plant hormones. The stem bud materials were inoculated separately with sterile water (control), *S. scitamineum* (inoculation), *S. scitamineum* supplemented IAA (inoculation + IAA), *S. scitamineum* supplemented GA$_3$ (inoculation + GA3) and *S. scitamineum* supplemented SA (inoculation + SA), and were collected at 0, 1, 3 and 7 d after inoculation. Error bars represent standard error of the mean (SEM). One way ANOVA was applied independently, and the least significant differences method was used for further comparison between two groups at $p < 0.05$ (marked with *) or $p < 0.01$ (marked with **).

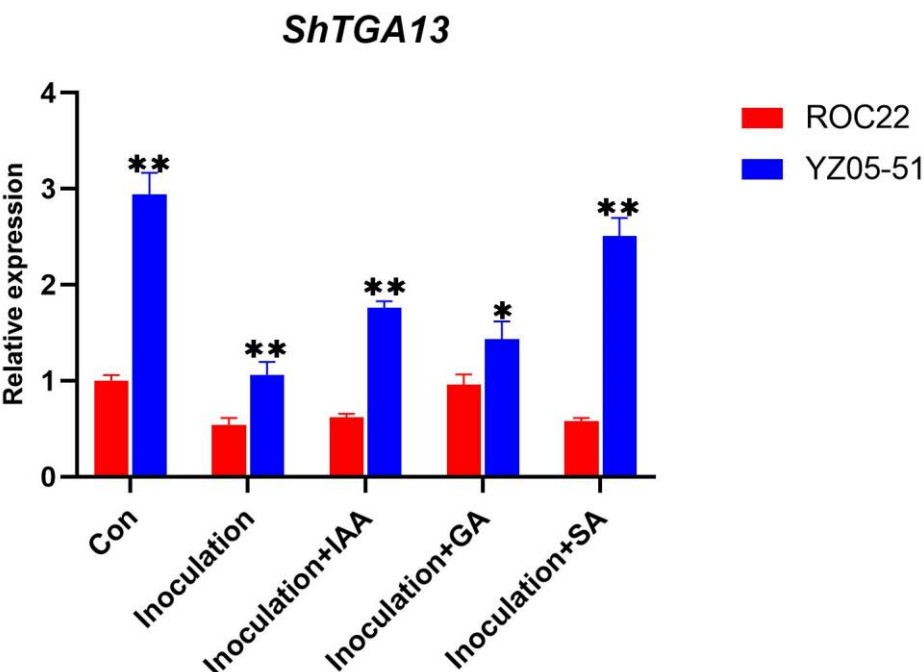

**Figure 5.** Relative expression of ShTGA13 gene in the susceptible and resistant varieties subsequent to *S. scitamineum* infection. The stem buds of ROC22 and YZ05-51 were inoculated separately with sterile water (control), *S. scitamineum* (inoculation), *S. scitamineum* supplemented IAA (inoculation + IAA), *S. scitamineum* supplemented GA$_3$ (inoculation + GA$_3$) and *S. scitamineum* supplemented SA (inoculation+ SA) and were collected at 7 d after inoculation. Error bars represent standard error of the mean (SEM). One way ANOVA was applied independently, and the least significant differences method was used for further comparison between two groups at $p < 0.05$ (marked with *) or $p < 0.01$ (marked with **).

## 4. Discussion

In plants, TGA transcription factors play crucial roles in regulating growth and development as well as pathogen defense. Nevertheless, there is nearly no report on the systematic study of the TGA family members in sugarcane. We used a genomic survey and identified sixteen ShTGA family members. The number of ShTGA members identified was greater than that of *Arabidopsis* (n = 10), tomato (n = 5) [57], peach (n = 15) [5], banana (n = 9) [6], *Taxus chinensis* (n = 12) [7], melon (n = 9) [8], but less than that of soybean (n = 25) [9]. It is possible that the larger number of *ShTGA* family genes was closely related to big genome size and divergence between the species. Variations of *TGA* family genes may provide a genetic basis for phenotypic variability. Analysis of conserved motifs and gene structure domains showed that most TGAs exhibited a similar pattern and a highly conserved DOG1 domain and bZIP domain, indicating that they shared similar functions. It was also present in other plant TGA transcription factors [5–8]. However, *ShTGA15/16* were distinct; these might be the possible pseudo-genes. Their proteins lacked the DOG1 domain, and the number of introns was far less than the other *ShTGA* genes. Furthermore, these two pseudo-genes were grouped into an outer group (Group VIII) in the phylogenetic analysis of *Arabidopsis*, rice and sugarcane TGA family proteins. Subcellular localization identified that ShTGAs are located in the nucleus, suggesting they mainly functioned in the nucleus.

Phylogenetic analysis demonstrated that sugarcane TGAs evolved differently from *Arabidopsis* TGAs, but showed an evolutionary close relationship with rice TGAs. In *Arabidopsis*, ten ATTGA family members were divided into five clades based on sequence homology. TGA1/4 comprised clade I, TGA2/5/6 belonged to clade II, TGA3/7 made up clade III, TGA9/10 were grouped into clade IV, and TGA8 (also named Pan) was separately grouped in clade V. While ShTGA transcription factor family members were evolutionarily

divided into eight groups, and the sizes of ShTGA family numbers in each divided group were obviously greater than those of ATTGA family numbers except for Group II, which was comprised of ATTGA1/3/4/7 and ShTGA1. This may indicate that sugarcane has maintained more diverse functions to better survive during evolution. It was initially considered that the AtTGAs from clades I, II and III mainly participated in the response to pathogen attack and abiotic stress, whereas AtTGA8/9/10 in clades IV and V were involved in regulating plant growth and development [4,42]. Nevertheless, an increasing number of reports show that most clades played roles in the regulation of defense response as well as developmental processes. For example, *Arabidopsis* TGA1 and TGA4 were found to be essential cofactors in the BLADE-ON-PETIOLE1(BOP)-dependent regulation required for SAM maintenance, flowering, and inflorescence architecture development [23]. ATTGA1/3/4/7 were found to function redundantly in the regulation of root growth and development, and TGA2/5/6/8 indeed had a common role in both promoting cell elongation and cellular redox photosynthesis [30]. The similar functions within ATTGA1/3/4/7 and ATTGA2/5/6/8 were consistent with our phylogenetical grouping. On the other hand, it was documented that clade IV AtTGAs played important roles in the defense of pathogens attack [58,59]. Similarly, it is possible that the functional division of *ShTGA* genes is not that evident in different groups, whereas functional redundancy is prevalent among the phylogenetically close genes in sugarcane.

It is well known that TGA are members of the bZIP transcription factors and regulate the transcription of the downstream genes to mediate a range of processes including the plant growth, development and adversity response [4]. The prediction of cis-regulatory elements in the promoter of *ShTGA* family genes supported these ideas that *ShTGA* genes were likely related to the responsiveness of hormone, adversity stress, light response, plant growth and development. In particular, a quantity of the hormone-responsive cis-regulatory elements in *ShTGAs* indicated that these genes may play roles in regulating resistance against pathogens in association with plant hormone signaling pathways. In this study, most test genes were involved in bud growth and developmental processes. Yet the interesting aspect here is that the levels of expression or expression patterns of *ShTGA13/14* were evidently altered when sugarcane was inoculated with *S. scitamineum*. It is highly likely that these genes were involved in the response to fungus infection. Phylogenetic analysis showed that ShTGA13 and OSbZIP49 were clustered into one group, and they had a close evolutionary relationship with ShTGA14. It was reported that OsbZIP49 regulated shoot growth and tiller via induction of the *indole-3-acetic acid-amido synthetase* genes (*GH3*) that catalyze the conjugation of auxins to amino acids as inactive forms to mediate local auxin homeostasis [60]. *OsbZIP49*-overexpressing transgenic rice exhibited an abnormal phenotype with increased tiller number, reduced plant height and internode lengths. Conversely, CRISPR/Cas9-mediated knockout of *OsbZIP49* displayed a compact architecture. After sugarcane suffered a pathogen attack, the expression of *ShTGA13* was depressed, which might decrease the activity of GH3 to convert auxin into inactive forms. It is well known that auxin is a vital virulence factor in some host–pathogen systems [61]. The *GH3* family genes were activators of plant disease resistance due to their functions in the regulation of IAA homeostasis in *Arabidopsis*, which was often associated with an SA-dependent pathway [43,62–64]. The rice *GH3* family could positively regulate bacterial and fungal pathogens by suppressing the loosening of the cell wall caused by auxin signaling [65,66]. *ShTGA13* was found to be highly expressed in the resistant variety compared to in the susceptible variety. These findings indicate that *ShTGA13* may also have positive role in regulating resistance to smut fungus in sugarcane, which is affected by pathogen attack and by adding IAA, GA and SA. It is expected that *ShTGA13* may be used for manipulating disease resistance to smut fungus in breeding sugarcane.

## 5. Conclusions

In this study, a total of sixteen *ShTGA* family genes were identified by way of a bioinformatic approach, most of which exhibited similar conserved motifs and contained a bZIP domain and a DOG1 domain. Phylogenetic analysis demonstrated that the sixteen ShTGA family members could be divided into eight clades, and evolved differently from *Arabidopsis* TGAs. All ShTGA family members suffered a purifying selection during evolution. A wide range of cis-regulatory elements was found in the promoter of *ShTGA* family genes including a hormone regulatory element, an adversity response element, a light responsive element, and growth and development regulatory elements. Most *ShTGA* genes were involved in bud growth and developmental processes except for *ShTGA10/11*. It is worth noting that the expression of ShTGA13/14 was depressed after sugarcane was infected with *S. scitamineum*, indicating that they may have roles in regulating resistance against *S. scitamineum*. Adding IAA, GA$_3$ and SA could restore the expression of *ShTGA13/14*, suggesting an association with a hormone-dependent regulatory pathway.

**Supplementary Materials:** The following supporting information can be downloaded at: https://www.mdpi.com/article/10.3390/agriculture12101644/s1, Table S1: The specific nucleotide sequences of primers for qRT-PCR; Table S2: The characteristics of conserved motifs in ShTGAs using MEME-suite; Table S3: The substitution rate ratio Ka/Ks and divergence time between the TGA gene family members.

**Author Contributions:** Conceptualization, C.W., Q.Z. and X.H.; methodology, Z.L., X.L. and X.H; validation, Z.W., X.L., C.W. and Q.Z.; formal analysis, Z.L., X.H. and Z.W.; resources, X.L., X.H., Z.W. and C.W; data curation, Z.L. and X.H.; writing—original draft preparation, Z.L. and X.H.; writing—review and editing, C.W., Q.Z. and Z.L.; supervision, C.W., Z.W. and Q.Z.; project administration, C.W., Q.Z. and X.H.; funding acquisition, C.W. and Q.Z. All authors have read and agreed to the published version of the manuscript.

**Funding:** This research was funded by China Agriculture Research System of MOF and MARA (grant number CARS-170101), Yunnan Fundamental Research Projects (grant number 2019FA016) and Science and Technology Mission of Cane Sugar Industry in Gengma County, Yunnan Province (grant number 202104BI090003).

**Data Availability Statement:** Not applicable.

**Acknowledgments:** Authors are grateful to the China National Germplasm Repository of Sugarcane for Supplementary Materials used for experiments. The authors also would like to thank Jiayong Liu, Bin Wu, Chunjia Li, Xiujuan Li, Ming Zhang, Yanhang Tang, Wei Zhang, Chenping Guan, Ying Li and Ziai Zhao for their assistance and consultations.

**Conflicts of Interest:** The authors declare no conflict of interest.

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
