# Peer review of "Identification and Expression Profiling of TGA Transcription Factor Genes in Sugarcane Reveals the Roles in Response to Sporisorium scitamineum Infection"

_agriculture, doi:10.3390/agriculture12101644_

Round 1

Reviewer 1 Report

Edited file is attached. There are few squestions/Suggestion for improvement

1- Intoduction part more information about already identified TGA and correlation with plant defense mechanism.

2- In material method 

a) After how many days of germination inoculation was performed?

b) DNA sequencing was perormed. But no data is shown in result section.

c) Details of qPCR are required in material and method section.

d) detail about cultivar SP80-3280 disease response? is it smut resistance??

e) why different conc of hormones are used , cite reference.

Result 

a) No sequencing of Genotype YZ05-51 TGAs. Do u think YZ05-51 TGAs are exactly the same like SP80-3280 .

b) No nucleotide related data is described in result section , length and homolgy etc.

c) Result and conclusion 

only change in expression of one ShTGA 13 without using or comparing with some susceptible variety is enough to support the role????

Author Response

Response to Reviewer 1' Comments

Thank you for your careful review. We really appreciate your efforts in reviewing our manuscript during this unprecedented and challenging time. We wish good health to you, your family, and community. Your careful review has helped to make our study clearer and more comprehensive. According to your advice, we amended the relevant part in manuscript. The main corrections are marked in red in the revised manuscript. Below is our response to your comments.

1- Intoduction part more information about already identified TGA and correlation with plant defense mechanism.

Response 1: Thanks for your good suggestion. We added the sentences “A amount of studies showed that the expression of TGAs was differentially regulated subsequent to pathogen infection, abiotic stress in plant [22-24]. Their regulatory roles were usually connected to plant hormonal pathways and they could affect plant immunity by modulating basal promoter activity of the PR-1 gene [25-26]” in the section of Intoduction.

2- In material method

  1. a) After how many days of germination inoculation was performed?

Response 2: Thanks for your good comments. Now we have revised the sentence “After three days of germination, sugarcane buds was injected with S.scitamineum inoculum via puncture as described previously [33]”.

  1. b) DNA sequencing was perormed. But no data is shown in result section.

Response 3: We are sorry for our incorrect writing. We didn’t perform DNA sequencing. We deleted “DNA sequencing” in the method.

  1. c) Details of qPCR are required in material and method section.

Response 4: Thank you for your comments. We add the sentence ”The volume of qRT-PCR reaction was 20 μL, including 10 μL FastStart Universal SYBR Green PCR Master, 0.4 μL primer and 2.0 μL template cDNA. qRT-PCR program was set at 95°C for 2min, 40 cycles of 95°C for 10 s and 60°C for 30 s.” in the section of method.

  1. d) detail about cultivar SP80-3280 disease response? is it smut resistance??

Response 5: The disease response of SP80-3280 is resistant. We have revised the sentence “The whole-genome shotgun contigs database of smut-resistant sugarcane cultivar SP80-3280 ....”.

  1. e) why different conc of hormones are used , cite reference.

Response 6: Thanks for your suggestion. We have revised the sentence “The specific phytohormones and their concentrations were 2 × 10−3 M IAA, 1 × 10−3 M gibberellin (GA3), 5 × 10−3 M SA referring to previous studies [45-46].”The cited references were also added in section of Reference.

Result

  1. a) No sequencing of Genotype YZ05-51 TGAs. Do u think YZ05-51 TGAs are exactly the same like SP80-3280.

Response 7: Good question. In my opinion, the sequences would not exactly the same between YZ05-51 and SP80-3280. We selected the same nucleotide sequences regions in both SP80-3280 and wild species AP85-441 to design specific primers for quantifying every ShTGA expression. The nucleotide sequences of primers should be also conserved in YZ05-51.

  1. b) No nucleotide related data is described in result section , length and homolgy etc.

Response 8: Thanks for your good comments. We added the sentences “The number of nucleotide of sixteen TGA genes was ranged from 2172 to 11238, and the sequence identity of their coding DNA sequences varied from 2.6% to 23.7%.” in the section of Result.

  1. c) Result and conclusion

only change in expression of one ShTGA 13 without using or comparing with some susceptible variety is enough to support the role????

α-Expansin EXPA4

Response 9: We are extremely grateful to reviewer for pointing out this problem. We have performed a supplementary experiment to compare the expression of ShTGA13 in susceptible (ROC22) and resistant (YZ05-51) varieties. The results showed that the expression of ShTGA13 was also decreased after ROC22 infected with S. scitamineum. It is noteworthy that ShTGA13 was significantly highly expressed in YZ05-51 compared to ROC22 under all experimental conditions. In general, the expression of ShTGA13 was regulated subsequent to pathogen infection, higher expression level was maintained in resistant variety indicating ShTGA13 had a relevance of resistance against S. scitamineum. Please refer to the revised paper and Figure 5 for the supplementary information.

Reviewer 2 Report

The manuscript is interesting since it identified 16 ShTGA family genes in sugarcane by a bioinformatic approach. Besides, a wide range of cis-regulatory elements in the promoter of these family genes including hormone regulatory element, adversity response element, light responsive element, and growth and development regulatory elements, were found. The expression of ShTGA13/14 was depressed after infecting sugarcane with S. scitamineum, revealing that they both may have roles in regulating resistance against this fungus. The addition of different hormones (IAA, GA3 and SA) restored the expression of ShTGA13/14 suggesting their association with hormone-dependent regulatory pathways.

The manuscript is in general well written, only minor syntaxes details are corrected in the file. Besides, the quality of some figures should be greatly improved.

Author Response

Response to Reviewer 2 Comments

Response: We appreciate the reviewer's positive evaluation of our work. We apologize for the mistakes in the manuscript and also carefully checked the entire manuscript for typographic, grammatical and formatting errors. We amended the relevant part in manuscript. According to your advice, we provided high quality of figures in supplementary files.

Reviewer 3 Report

The authors report drastic reduction in expression of ShTGA13/14 genes after S. scitamineum inoculation in sugarcane and claim that they may have roles in regulating resistance against S. scitamineum. This is not convincing.

The authors have inoculated the pathogen only in a resistant variety for gene expression studies. The results can be more authentic if gene expression studies were conducted in a set of resistant and susceptible varieties.  

The presented data do not support the hypothesis on the role of TGA transcription factor genes in sugarcane Sporisorium scitamineum resistance.

Author Response

Response to Reviewer 3 Comments

Special thanks for your good suggestions and valuable comments. We acknowledge the reviewer's comments and suggestions very much, which are valuable in improving the quality of our manuscript.

Response 1: We are extremely grateful to reviewer for pointing out this problem. We have performed a supplementary experiment to compare the expression of ShTGA13 in susceptible (ROC22) and resistant (YZ05-51) varieties. The results showed that the expression of ShTGA13 was also decreased after ROC22 infected with S. scitamineum. It is noteworthy that ShTGA13 was significantly highly expressed in YZ05-51 compared to ROC22 under all experimental conditions. In general, the expression of ShTGA13 was regulated subsequent to pathogen infection, higher expression level was maintained in resistant variety indicating ShTGA13 had a relevance of resistance against S. scitamineum. Please refer to the revised paper and Figure 5 for the supplementary information.

Round 2

Reviewer 1 Report

After required changes,  its now ok.

Just recheck reference citation.